# Herd Immunity to the Measles, Mumps and Rubella Viruses Among the Belgradian Population in May, 2024

**DOI:** 10.3390/vaccines13060652

**Published:** 2025-06-18

**Authors:** Anna Y. Popova, Vyacheslav S. Smirnov, Svetlana A. Egorova, Luka Dragačević, Angelica M. Milichkina, Jelena Protić, Ekaterina M. Danilova, Irina V. Drozd, Marija Petrušić, Ojuna B. Zhimbaeva, Elizaveta S. Glazkova, Nataša Gutić, Valeri A. Ivanov, Edward S. Ramsay, Oleg V. Kotsar, Vyacheslav Y. Smolensky, Areg A. Totolian

**Affiliations:** 1Federal Service for the Oversight of Consumer Protection and Welfare,127994 Moscow, Russia; 2Saint Petersburg Pasteur Institute, 197101 St. Petersburg, Russia; vssmi@mail.ru (V.S.S.); zhimbaeva@pasteurorg.ru (O.B.Z.); totolian@spbraaci.ru (A.A.T.); 3Institute of Virology, Vaccines, and Sera “Torlak”, 11152 Belgrade, Serbia; ldragacevic@torlak.rs (L.D.); jprotic@torlak.rs (J.P.); mpetrusic@torlak.rs (M.P.); ngutic@torlak.rs (N.G.)

**Keywords:** measles, mumps, rubella, Belgrade, seropositivity, vaccination, specific IgG antibodies, MMR vaccines

## Abstract

**Background/Objectives:** In the Republic of Serbia, measles vaccination was first introduced in 1971, while combined vaccination (measles, mumps, rubella) was made mandatory in 1996 as part of the national vaccination program. Reported prevalence values for 2023 were <0.75 cases per 100K population for measles, 0.09 cases per 100K for mumps, and no cases of rubella. **Methods**: This cross-sectional study was performed in May, 2024 as part of the project “Herd Immunity to Vaccine-Preventable and Other Relevant Infections in the Belgradian Population.” It focused on assessing herd immunity to measles, mumps and rubella (MMR) among residents insofar as these remain a public concern despite the availability of vaccines. A total of 2533 subjects were distributed across nine age groups, covering those aged 1–70^+^ years and various professional groups residing in Belgrade. Participants were stratified by age and activity. Upon obtaining individual information by online questionnaire and receiving a signed statement of informed consent, blood samples were obtained for IgG antibody testing (ELISA) to determine MMR serological status. The results were compared to national and international immunization standards to evaluate herd immunity levels. **Results**: Our results indicate varying levels of immunity for each virus, with specific demographic groups showing different immunity levels. Total measles seroprevalence during this study was 74.7%, with significant variation across all age groups. While high seropositivity was observed in both children (90.7%) and elder age groups (98.4%), middle-aged individuals in the age group 30–49 years showed significantly lower IgG levels. Between 2021 and 2023, there were no registered cases of rubella detected in Serbia, which indicates a high level of immunity. This was confirmed here with consistently high IgG levels across all age groups, with an average seropositivity of 94.8%. Average mumps seropositivity across all age groups was 85.1%. The lowest value was in the young child (1–5 years) age group (76.1%); the highest was in the elderly group (92.6%). **Conclusions**: The current findings suggest that the Belgradian population has strong overall immunity to MMR, yet with some concerns regarding measles immunity in middle-aged adults, suggesting a potential need for catch-up vaccinations. While rubella status indicates strong herd immunity and minimal risk of outbreaks, mumps immunity in some groups (children, middle-aged adults) is below the protective threshold. While it is still sufficient to prevent widespread transmission, it should be closely observed. To our knowledge, this study is the first of its kind to provide data about MMR seroprevalence in Belgrade. Findings indicate the need for constant surveillance and revaccination of vulnerable/seronegative groups.

## 1. Introduction

Despite the impressive successes of modern medicine, measles, mumps, and rubella (MMR) have not lost their relevance and still pose a serious public health problem. Insofar as targeted therapies for MMR viral illnesses have not yet been developed, vaccines remain the only possible means of prevention. Separate monovalent vaccines against each virus and preparations containing two or three attenuated MMR viruses in various combinations have been created and are used in medical practice. Only one such combination vaccine, the trivalent design, elicits immunity to all MMR viruses at once. Three-component vaccines have received the widest distribution globally. They contain various MMR strains and are marketed under different brand names: Priorix, M-M-R II, Vactrivir, Trimovax, etc. [1,2,3,4,5,6,7,8].

The widespread use of highly effective MMR vaccines served as a prerequisite for the WHO to set a goal of completely eliminating MMR in most countries [9]. However, despite some successes, the desired progress in solving this problem has not yet been achieved, as outbreaks of MMR infections are observed annually in both developing and industrialized countries [10,11,12,13]. Since 2018, serious measles outbreaks have been reported in Europe, including Poland [14], Iceland [15], Bosnia and Herzegovina [16] and several other European countries [17,18,19,20].

According to the UN, for the period from January to October 2023, forty member states reported 30,000 cases of measles, which is 30 times higher than for all of 2022 [20]. According to an analysis by the WHO and the United Nations Children’s Fund (UNICEF), 127,350 measles cases were reported in the European Region for 2024 [21]. This is double the number of cases reported for 2023 and the highest number since 1997. These data are alarming regarding the emerging epidemiological situation, especially in light of the risk for potential complications (acute and chronic) that measles infection bears.

Vulnerability to measles and mumps is often due to negative socioeconomic factors, such as poverty, overcrowding, insufficient access to healthcare, etc. [22,23,24]. In addition, increased susceptibility to MMR infections may be due to environmental and biological causes, including unfavorable climatic factors, other viral infections (such as HIV or COVID), intoxications, tissue and organ transplants, neoplastic diseases, etc. [25,26,27,28,29,30,31]. In the European continent, a key cause of the measles outbreak is low vaccination compliance in the young. This is confirmed by information from the European Centre for Disease Prevention and Control that eight out of ten people diagnosed with measles in the EU/EEA in the last year were unvaccinated [32].

In the Republic of Serbia, the national vaccination schedule indicates the first dose of MMR vaccine to be administrated when the child turns 12 months of age (second year), and the second dose should be administrated before the first grade (no later than seven full years of age). Obligatory MMR immunization of children became a part of the national immunization program in 1993, and the MMR vaccine has been given in two doses since 1996 [33]. Earlier (before 1993), monovalent mumps and measles vaccines were administrated.

The situation with mumps and rubella in Europe has been less concerning in previous years. However, individual cases were registered until recently, with mumps being the most frequently registered [34]. In Serbia, the epidemiological situation for the period from 2021 to 2023 was quite favorable [35]. In 2021, 1 case of measles and 4 cases of mumps were registered. In 2022, no patients with measles were identified, and mumps was diagnosed in 11 people. In 2023, the number of patients with measles was 50 (0.74 per 100K pop.), and there were six mumps cases. No cases of rubella were registered during this period.

Since 1992, the Dutch MMR vaccine ‘M-M-RVaxPro^®^ (Merck Sharp & Dohme B.V., J07BD52) has been used in Serbia. It is a live, attenuated design. According to available data, vaccination coverage has been 84.1–91% in recent years (Table 1) [35].

The data in Table 1 shows the achievement of the required vaccination coverage in the Serbian population in 2023. Unfortunately, we were unable to obtain similar data regarding 2023 for the Belgradian population. However, available results for 2022, indicating the achievement of 83% coverage by planned vaccination, may indicate a high, although not absolute, level of protection of the capital’s population against measles and rubella.

A more accurate judgment regarding MMR seropositivity levels can be made based on herd immunity survey results. The levels of measles and rubella seropositivity in the Serbian population were 76.2% (95% CI: 73.4–78.8) and 86.1% (95% CI: 83.8–88.2) in 2018, respectively [13]. Similar data regarding mumps herd immunity in the Serbian population could not be found. Likewise, no information was found on herd immunity in the Belgradian population. Studies about MMR seroprevalence are limited, and the few that are published in peer-reviewed journals reflect the epidemiological situation in the northern Serbian province.

In a study on Vojvodina Province (Serbia) published in 2019, it was reported that 86.9% of serum samples were seropositive for measles (median age was 20 years) [36]. The highest share of measles seronegativity was observed in children aged 12–23 months and in adults aged 20–39 years (56.1% and 18.5%, respectively). A different study examined the percentage of participants seropositive for anti-rubella antibodies, which was 92.9% in the entire sample (Vojvodina region only) [37]. The highest seronegativity was in the youngest (1 year) age group (44.7%), followed by the groups aged 24–49 (6.4%) and 2–11 years (6.2%).

To our knowledge, there are no published studies on MMR seroprevalence in the Belgrade region. This situation prompted the planning and execution of the current study. The seropositivity levels for MMR viral pathogens were assessed, as well as vaccination coverage among the population of Belgrade.

## 2. Materials and Methods

### 2.1. Description of the Volunteer Cohort

A cross-sectional randomized study, “Herd immunity to vaccine-preventable and other relevant infections in the Belgradian population”, was conducted in May, 2024 under a joint program between Rospotrebnadzor (Russia) and the Serbian Ministry of Health. It was approved by the relevant local ethics committees at the Saint Petersburg Pasteur Institute (Saint Petersburg, Russia) and the Institute of Virology, Vaccines, and Sera “Torlak” (Belgrade, Serbia). Before the study, all participants, or their legal representatives, were familiar with the purpose and methodology of the study and signed a statement of informed consent. The selection of volunteers was carried out using a web application (Saint Petersburg Pasteur Institute, Saint Petersburg, Russia). The size of a representative sample was calculated using the previously described method [38,39]. The total number of volunteers examined in the cohort was 2533. The cohort was stratified by age and gender, as shown in Table 2.

As follows from the data, the cohort was dominated by middle-aged volunteers in the range of 30 to 59 years old, whose total share was 65.4% (95% CI: 63.5–67.2). In terms of gender, the cohort included 67.7% females and 32.3% males; the number of females in the cohort was 2-fold higher than males (Table 2). The largest percentages of women were noted in the age groups 18–29 and 40–49 years (75.5% and 70.9%, respectively). The distribution of volunteers by activity is presented in Table 3.

The most active participants in the research were healthcare workers (21.4%) and pensioners (14.3%) (Table 3). The lowest number were military personnel and schoolchildren. To increase representativeness during data analysis, military personnel were combined with civil servants, and schoolchildren were combined with preschoolers.

### 2.2. Research Methods

#### 2.2.1. Volunteer Inclusion Criteria

During a broad information campaign, individuals filled out an online questionnaire via a web application, including personal data: full name, gender, age, area of residence, field of activity, chronic disease status, and contact information. Individuals who met the inclusion criteria (no acute illnesses at the time of the study, not on immunosuppressive therapies) were invited to have their blood drawn and undergo subsequent laboratory testing. The methodology for organizing and conducting the study has been described in detail earlier [39]. At the blood collection office, the registrar, together with the volunteer, filled out an extended questionnaire. This included questions about past MMR illness and other vaccine-preventable infections. Information on immunization (vaccination, revaccination) against the listed infections was also collected, including vaccine names and dates of usage. Information about previous MMR illness history was obtained from the volunteer verbally. Information about vaccination was obtained from certificates provided by the volunteer (if available) and clarified from other medical records. In the absence of such records, verbal statements from volunteers were used. The collection of anamnestic data and blood samples lasted for two full weeks in May, 2024.

#### 2.2.2. Sample Collection and Testing

For subsequent laboratory detection of specific IgG antibodies, blood samples were taken from the cubital vein of each volunteer into vacutainers with a solution of ethylenediaminetetraacetic acid (K_3_EDTA). After centrifugation, blood plasma was separated from cellular elements, transferred to microtubes, and stored at 4 °C until testing. ELISA analyses were performed using commercial reagent kits (Vector-Best JSC, Novosibirsk, Russia) according to manufacturer instructions: “VectoMeasles-IgG” for the presence and level (IU/mL) of anti-measles IgG; “VectoRubella-IgG” for the presence and level (IU/mL) of anti-rubella IgG; and “VectoParotit-IgG” for the presence of anti-mumps IgG.

The presence and titers of anti-measles and anti-rubella IgG antibodies were determined using registered Russian reagent kits approved by the WHO for monitoring measles and rubella immunity. Each reagent set includes calibration samples with known IgG concentrations (against measles virus—0, 0.15, 0.5, 1, 2, 5 IU/mL; against rubella virus—0, 10, 50, 100, 200, 800 IU/mL) and one control sample of known IgG concentration. The average optical density values for each pair of wells by type (calibration, control, or test sample) were calculated. Next, the calibration curve (optical density versus IgG concentration in calibration samples) was plotted. The concentrations of anti-measles and anti-rubella IgG in the control and test samples were then calculated using the calibration curve”.

For anti-measles IgG, results were interpreted as follows: positive (≥0.18 IU/mL), negative (<0.12 IU/mL), or inconclusive (0.12–0.17 IU/mL). For anti-rubella IgG, results were interpreted as follows: positive (≥10 IU/mL) or negative (<10 IU/mL).

For anti-mumps IgG, a cut-off optical density (OD_crit._ = OD_media for K−_ + 0.3) and a positivity coefficient (PC = OD_sample_/OD_crit._) were calculated. The results were interpreted as follows: positive (PC > 1), negative (PC < 0.8), or inconclusive (0.8 ≤ PC ≤ 1).

### 2.3. Statistical Processing

Statistical processing was performed by methods of variation statistics using Excel 2011. Statistical processing of proportions was performed using the method of A. Wald and J. Wolfowitz [40], as modified by A. Agresti and B.A. Coull [41]. The significance of differences in proportions was calculated using the z-test [42]. Differences were considered significant at *p* ≤ 0.05 unless otherwise stated.

## 3. Results

### 3.1. Measles

During serological testing, the percentage of individuals with anti-measles IgG was 74.7% (95% CI: 73.0–76.4), with uneven distribution across the cohort (Figure 1).

The highest seropositivity values were found in children aged 6–11 years (90.7%; 95% CI: 77.8–96.9), adults aged 60–69 years (98.4%; 95% CI: 96.3–99.4), and those aged 70^+^ years (98.9%; 95% CI: 96.0–100). Significantly lower measles seropositivity was noted among middle-aged volunteers (30–49 years): 57.2% (95% CI: 54.4–60.0). All differences were significant relative to the overall cohort average.

According to the literature, 95% is the protective level [43]. Therefore, seroprevalence reached the protective threshold in older age groups. In children 1 to 17 years old, seroprevalence almost reached the protective value. As such, only among middle-aged individuals (30–49 years) would a measles catch-up vaccination potentially be required. In light of the noted age group differences, seroprevalence depending on activity was also analyzed for potential differences and their significance.

When assessing seroprevalence by activity, protective levels were found among children (preschoolers, schoolchildren) and pensioners, alongside lower levels in the ‘scientist’ and ‘other’ groups (Figure 2).

Considering that most people in these categories had clearly reached middle age at the time of the survey, it can be assumed that the identified seroprevalence dependencies in different activity groups are in satisfactory agreement with the seroprevalence distribution by age.

In addition to calculating seroprevalence, the study also quantitatively assessed anti-measles IgG levels in volunteers of different ages (Figure 3).

These data indicate a pronounced heterogeneity in anti-measles IgG seroprevalence. The distribution of seronegative volunteers (<0.18 IU/mL) indicates a low percentage of such individuals aged 1 to 29 years. The largest number was noted in the group aged 30–49 years, which is consistent with the age-related seroprevalence. As volunteer age increased, the share seronegative progressively decreased to almost zero. Regarding the distribution of weakly or moderately seropositive subjects (0.18–0.5, 0.51–1.0 IU/mL), most were detected among children (1–11 years old). A gradual decrease in the representation of such volunteers to 1.1% by the age of 70^+^ years was noted.

The greatest differences in Figure 3 were noted across age groups in the form of the trend line for volunteers with maximum IgG levels (>2 IU/mL). The group ‘children aged 1–5 years’ did not have any such individuals. In those aged 6–11 years, the share with such IgG levels increased to 16.3%, in the context of 90.7% seroprevalence (Figure 1) for the age group. In age groups spanning 12–39 years, a gradual decrease to 4.4% was seen. Subsequent growth to 12.8% (40–49 years) was noted, followed by an almost exponential growth to the maximum level of 75.1–76.6%. Such growth was accompanied by an increase in seroprevalence to 98.4–98.9% among individuals aged 60–70^+^ years (Figure 1).

### 3.2. Rubella

The epidemiological situation regarding rubella in Serbia remains favorable overall. According to the official data of the Institute of Virology, Vaccines and Sera “Torlak”, no cases of rubella were detected in Serbia in the period from 2021 to 2023.

Stratification of volunteers by age and activity did not reveal any significant seroprevalence differences in the Belgrade population (Figure 4).

Seropositivity was 94.7% (95% CI: 93.8–95.6) on average, with all age and professional groups of respondents falling within its confidence interval.

The distribution of IgG levels by age showed a number of specific features (Figure 5).

First of all, the low share of seronegative individuals in all age categories is noteworthy. The shares of individuals with low IgG levels (10.1–25.0 IU/mL) are distributed similarly. In this context, the distribution of very high IgG levels deserves special attention. In children aged 1–17 years, the share of individuals with such level varied from 15.4 to 19.4%. However, starting with volunteers aged 18–29 years and older, a sharp increase in the share of seropositive individuals with very high IgG levels is observed. The maximum occurred in volunteers aged 60–69 years, amounting to 60.9% (95% CI: 55.5–66.1). It is interesting to note that the opposite trend was observed for volunteers with IgG levels in the range of 25.1–100.0 IU/mL.

Anti-rubella IgG levels were compared in volunteers who had experienced rubella and those who had been vaccinated, as follows. The former subset (n = 43) confidently indicated past illness (1965–1997). The latter subset (n = 85) confirmed actual vaccination (1977–2024) with medical documentation and denied any history of illness in the anamnesis. No reliable differences were noted in the levels of post-infectious and post-vaccination rubella immunity. Most volunteers were in the high IgG level category ( > 100 IU/mL): 90.7% (95% CI: 77.9–97.4) among those who ‘had experienced illness’; and 68.3% (95% CI: 57.1–78.1) among those who ‘had been vaccinated’. Seronegative status did not exceed 2.5% in either group. Interestingly, among volunteers who confidently denied any history of illness or vaccination (386 people >30 years old), seroprevalence (92.7%; 95% CI: 89.7–94.9) was consistent with the cohort, with more than 70% having high IgG levels >100 IU/mL.

Thus, the obtained data indicate the protective seroprevalence level among the Belgradian population.

### 3.3. Mumps

In the 2021–2023 period, the epidemiological situation regarding mumps in Serbia was characterized by sporadic cases (Table 4). In this context, a similar situation would be expected in the Republic’s largest city, Belgrade.

The most likely reason for this situation may be a high level of mumps herd immunity. According to the results, seropositivity in the cohort averaged 85.1% (95% CI: 83.7–86.4). Analysis of the age distribution of seropositivity showed that in the cohort, it varied from 76.9% (95% CI: 49.7–91.8) in children aged 1–5 years to 92.6% (95% CI: 88.0–95.5) in those aged 70^+^ persons. In almost all groups, differences were insignificant, only reaching significance with the 70^+^ group (*p* < 0.05) (Figure 6).

The distribution of seropositivity in relation to volunteer activity was in good agreement with the distribution by age. A significant difference was found only in the “pensioner” group (91.7%; 95% CI: 88.4–94.1), which, obviously, includes mostly elderly individuals. In all other groups, differences were insignificant.

### 3.4. Vaccination Status

Unfortunately, we were unable to obtain complete and reliable information on the types of vaccines used. Of the volunteers included in the study (n = 2533), the following reported vaccination: against measles—1473 people (56.1%; 95% CI: 54.2–58.0); against rubella—1258 people (47.9%; 95% CI: 46.0–49.8) and against mumps—1302 people (49.6%; 95% CI: 47.7–51.5). As expected, vaccination coverage for the three infections was roughly the same. It decreased with age from 95% in children’s groups to 12–15% among volunteers over 60 years old (Figure 7).

In over 70% of cases, volunteers did not provide vaccination data. Vaccination information confirmed by medical documentation (Table 5) was provided for measles (112 individuals), rubella (85 individuals), and mumps (91 individuals). According to the National Drug Registry (NRL 2024, https://registar.alims.gov.rs/ (accessed on 1 March 2025), the main vaccine used in Serbia was a three-component (MMR) live vaccine (various manufacturers).

## 4. Discussion

We aimed here to provide, to our knowledge, the first of its kind: a comprehensive, cross-sectional evaluation (May 2024) of measles, mumps, and rubella (MMR) seroprevalence among the Belgradian population. As such, it offers insight into both immunity levels and potential areas of concern.

### 4.1. Measles Immunity: Potential Need for Catch-Up Vaccination

The measles virus is extremely contagious, and a very high level of herd immunity is required for successful public health control [44]. Overall, measles seroprevalence in Belgrade was found to be 74.7%, which is below the widely accepted protective threshold of 95% necessary for herd immunity. High seropositivity levels were observed among children aged 6–11 years (90.7%) and older adults (98.4%). Middle-aged adults (30–49 years) had significantly lower immunity (57.2%). This age group likely featured individuals who either missed vaccination, did not receive booster doses, or experienced waning immunity over time. Similar trends have been reported in European countries, where gaps in measles immunity among adults have been linked to insufficient past measures [32,45].

The high share of seropositive volunteers with high IgG levels among preschool children is most likely due to the recent revaccination of children in this age group. As for elderly and older individuals, high IgG levels in the absence of vaccination indicate the presence of an anamnestic response to childhood illness. Analysis of IgG levels among people who have experienced illness versus those vaccinated confirms the duration and intensity of post-infectious measles immunity compared to the post-vaccination response.

The study data on measles herd immunity in Belgradian residents show a high level of seroprevalence, especially among preschool children, the elderly, and older volunteers. Caution is needed when interpreting the findings in the preschool group as its sample size was rather small. Some concern is caused by seroprevalence levels in the most active middle-aged population categories. A significant share of them had not reached the protective threshold. This may be due to deviations from the methodology of measles preventive immunization.

### 4.2. Rubella: Sustained Immunity and Low Outbreak Risk

The study confirmed a high level of rubella seropositivity (94.8%) across all age groups. This is consistent with the absence of rubella cases in Serbia between 2021 and 2023 [35], which is due to the successful achievement of herd immunity. These results suggest that the current immunization strategy has been effective in preventing rubella transmission, which supports the country’s efforts toward rubella elimination according to the WHO Immunization Agenda 2030 [46].

Notably, the highest IgG levels were observed in older individuals (60–69 years), likely reflecting past exposure to the virus before widespread vaccination was implemented. In contrast, younger individuals, particularly those in the ‘1–17 year’ age group, exhibited lower (but still sufficient) IgG levels. This reinforces the role of routine childhood vaccination in maintaining high collective immunity. A gradual decline in vaccinated individuals across advancing age groups is expected, keeping in mind the initiation of full MMR vaccination in the former Yugoslavia (1993).

### 4.3. Mumps Immunity: Areas for Improvement

Mumps seroprevalence was found to be 85.1%. The lowest was observed in children aged 1–5 years (76.1%), and the highest was seen in older adults (92.6%). This is below the accepted 95% protective threshold [47,48]. However, judging by official data [35], it is sufficient to prevent the widespread spread of mumps in the Republic, although isolated cases are still possible. Given that mumps outbreaks have occurred in vaccinated populations due to waning immunity [49], continued surveillance is necessary.

In this study, mumps serological status was assessed qualitatively, which does not allow comparison of IgG levels produced in response to infection or vaccination.

### 4.4. Conclusions Common to the Three Infections

It is noteworthy that seroprevalence among middle-aged and older individuals who confidently denied any history of illness or vaccination (i.e., presumably had no contact with the viral pathogen) was high. Such a “discrepancy” is present for all infectious agents in this study. This may be due to the fact that elderly individuals frequently do not remember being sick in early childhood, or in some rare cases, they may have experienced asymptomatic infection [50]. This assumption is supported by high IgG levels in most of these volunteers.

The overall findings emphasize the critical role of maintaining high vaccination coverage to prevent outbreaks of vaccine-preventable diseases. In the Republic of Serbia, anti-measles vaccination was first introduced in 1971. Combined vaccination (MMR) was made mandatory in 1996 as part of the national vaccination program. In this context, the vaccination coverage of Belgradian residents who took part in the study was similar for the three infections and, as expected, decreased with age from 95% in children’s groups to 12–15% among volunteers over 60 years of age.

While rubella immunity appears sufficient, measles and mumps require additional efforts to close immunity gaps, particularly among middle-aged adults and young children. We have identified this situation not only in Serbia but also in other regions where similar studies were conducted in 2023: in St. Petersburg and the Leningrad Region in the Russian Federation [51], as well as in the Kyrgyz Republic [52]. Many middle-aged adults are unaware of their immunization status, and to address these issues, the following public health measures should be considered: (i) catch-up vaccination campaigns targeting adults with low measles immunity and (ii) increased surveillance and monitoring of mumps immunity trends, particularly in children and middle-aged individuals.

The situation with individual seroprevalence differences (measles, mumps, rubella) in the context of trivalent MMR vaccine usage can be explained by several factors. The first may be differences in the immunogenicity of individual vaccine components. The most immunogenic component is apparently the attenuated rubella virus, which, according to the survey data, causes an immune response in more than 90% of those vaccinated. A second factor may be differences in the stability and duration of post-vaccination immunity. It may be that with enough time following vaccination, small differences become apparent. Regarding measles, the lowest seroprevalence was indeed noted in persons aged 18 to 49 years, that is, 10 or more years after vaccination.

## 5. Conclusions

1. Our findings suggest the need for targeted catch-up measles vaccination programs for middle-aged adults. Given that measles outbreaks have been increasingly reported across Europe, especially in unvaccinated individuals, ensuring adequate immunity in this age group is essential to prevent transmission and potential outbreaks in Belgrade.

2. The obtained data reflect rubella seroprevalence values almost reaching the protective level in the Belgradian population. This, together with the data for Serbia as a whole, indicates the absence of epidemiological prerequisites for the spread of rubella in the Republic, including its capital, Belgrade.

3. Continued surveillance for mumps serological status is necessary.

4. This study provides valuable epidemiological data for guiding immunization policies. While Belgrade’s population exhibits strong overall immunity to MMR, targeted interventions are required to address specific gaps in measles and mumps immunity. Maintaining high vaccination coverage and considering booster doses for vulnerable age groups will be crucial in achieving sustained protection against these diseases.

## 6. Limitations of the Study

The authors would like to note several factors that might affect sample representativeness or conclusions reached through data analysis. Women are more involved with their health and that of their loved ones, and they are more likely to take part in these studies. The predominance of pensioners and health workers in the cohort may be due to the higher social activity and more responsible attitude towards the health of volunteers in these two categories.

Caution is needed when interpreting the findings in the children groups (especially preschool) due to the small sample size.

Initially, information about history (illness, vaccination) was taken from volunteer oral statements and from vaccination certificates they provided. The authors understand that for adult participants, there is a high probability that the volunteer may not remember prior illness or vaccination. In most cases, these data could not be verified using medical records.

## Figures and Tables

**Figure 1 vaccines-13-00652-f001:**
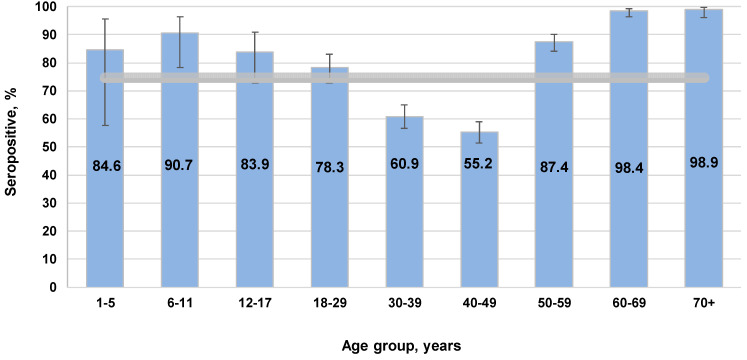
Measles seropositivity (IgG) by age. Here and in Figure 2, Figure 3, Figure 4, Figure 5, Figure 6 and Figure 7, vertical black lines are confidence intervals. The overall cohort value (horizontal band) was 74.7% (95% CI: 73.0–76.4).

**Figure 2 vaccines-13-00652-f002:**
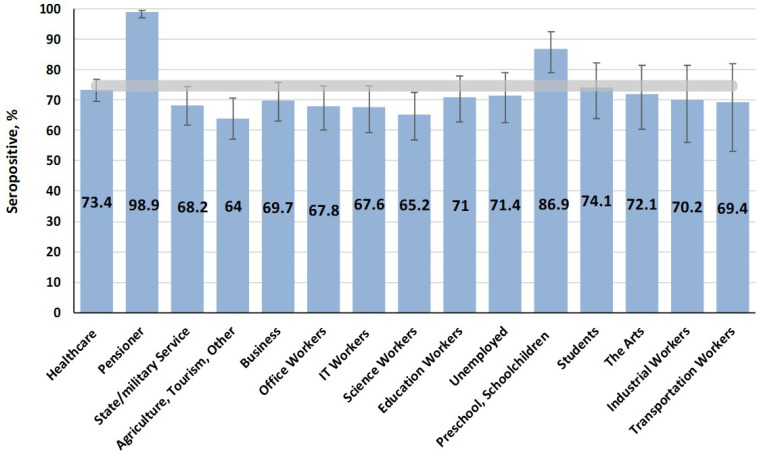
Measles seropositivity (IgG) by activity. The overall cohort value (horizontal band) was 74.7% (95% CI: 73.0–76.4).

**Figure 3 vaccines-13-00652-f003:**
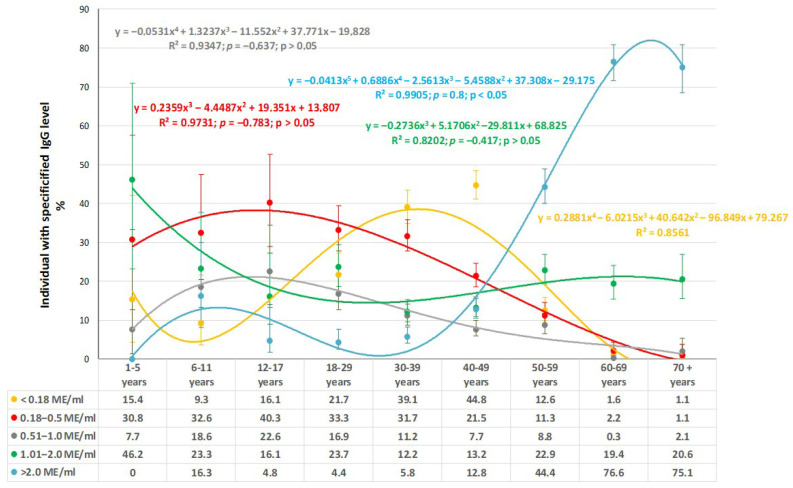
Quantitative anti-measles IgG level trends by age (IU/mL). Note: regression equations, determination coefficients (R^2^), correlation coefficient values (ρ), and *p* values are shown at the top (in matching trend colors).

**Figure 4 vaccines-13-00652-f004:**
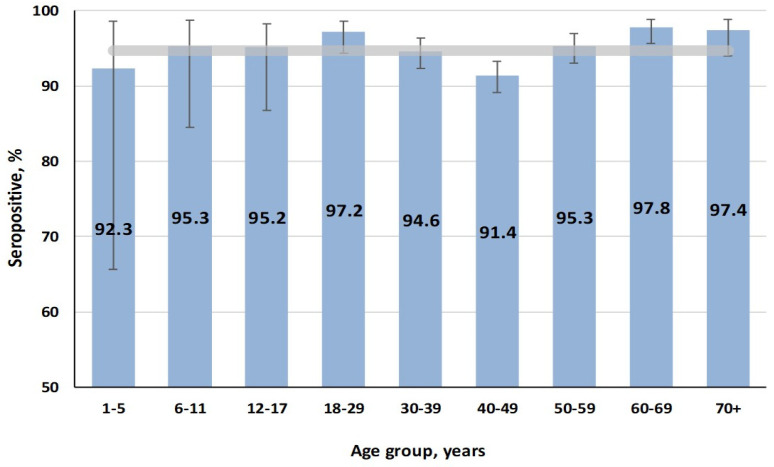
Rubella seropositivity (IgG) by age. The overall cohort value (horizontal band) was 94.7% (95% CI: 93.8–95.6).

**Figure 5 vaccines-13-00652-f005:**
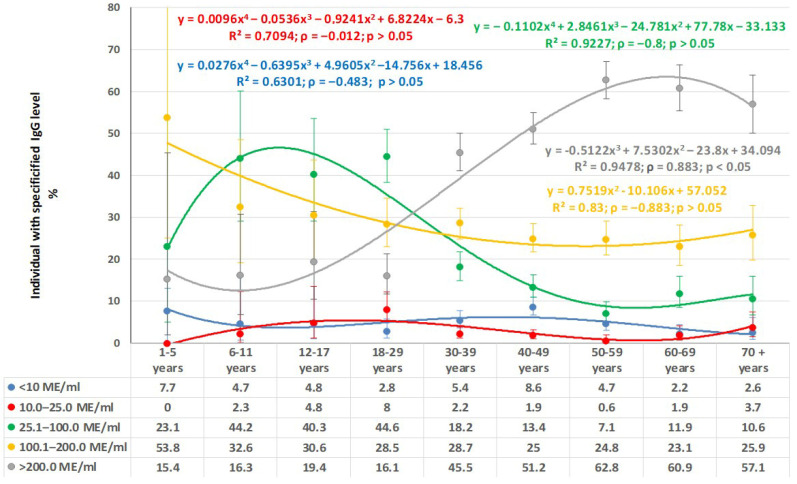
Quantitative anti-rubella IgG level trends by age (IU/mL). Notes: regression equations, determination coefficients (R^2^), correlation coefficient values (*p*), and *p* values are shown at the top (in matching trend colors).

**Figure 6 vaccines-13-00652-f006:**
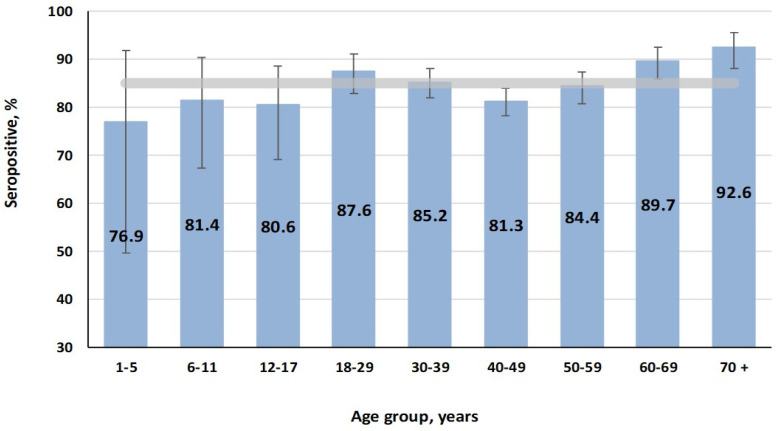
Mumps seropositivity (IgG) by age. The final cohort value (horizontal band) was 85.1% (95% CI: 83.7–86.4).

**Figure 7 vaccines-13-00652-f007:**
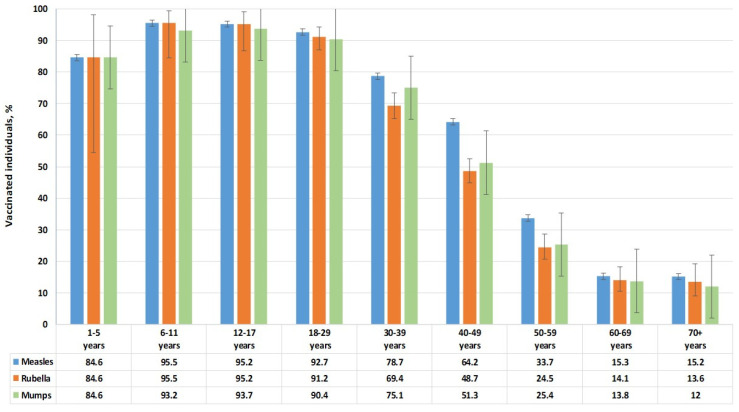
Vaccination coverage of volunteers surveyed in Belgrade.

**Table 1 vaccines-13-00652-t001:** Vaccination coverage of the Serbian population, 2020–2023.

Year	Vaccination Coverage (M-M-RVaxPro)	
	Initial Immunization ^a^, %	Revaccination ^b^, %
2020	78.1	84.1
2021	74.8	85.8
2022	81.3	89.5
2023	84.5	91.0

Notes: data provided by the Institute of Public Health of Serbia “Dr. Milan Jovanović Batut”; (a) immunization at age ≥ 2 years; (b) revaccination after six full, and before seven years of age.

**Table 2 vaccines-13-00652-t002:** Demographic structure of the cohort.

Age Group, Years	Individuals, N	Share, %	95% CI	Males		Females	
				n	%	n	%
1–5	13	0.5	0.3–0.9	6	46.2	7	53.8
6–11	43	1.7	1.2–2.3	25	58.1	18	41.9
12–17	62	2.4	1.9–3.1	35	56.4	27	43.6
18–29	249	9.8	8.7–11.0	61	24.5	188	75.5
30–39	501	19.8	18.3–21.4	161	32.1	340	67.9
40–49	688	27.2	25.5–28.9	200	29.1	488	70.9
50–59	468	18.5	17.0–20.0	151	32.3	317	67.7
60–69	320	12.6	9.4–14.0	104	32.5	216	67.5
70^+^	189	7.5	6.5–8.6	76	40.2	113	59.8
Total	2533	100.0	-	819	32.3	1714	67.7

Notes: N—age group size; n—number male or female; 70^+^ is persons aged ≥ 70 years; CI—confidence interval.

**Table 3 vaccines-13-00652-t003:** Cohort description by activity.

Activity	Individuals, N	Share, %
Healthcare Workers	541	21.4
Pensioners	363	14.3
State-military Service	209	8.3
Other	200	7.9
Business	198	7.8
Office Workers	152	6.0
Internet Technology	139	5.5
Science	138	5.4
Education	138	5.4
Unemployed	112	4.4
Preschoolers/Schoolchildren	107	4.2
Students	85	3.4
The Arts	68	2.7
Industrial Workers	47	1.9
Transportation workers	36	1.4

**Table 4 vaccines-13-00652-t004:** Mumps incidence among the Serbian population, 2021–2023.

Year	Age Group, Years									Total
	1–4	5–9	10–14	15–19	20–29	30–39	40–49	50–59	60^+^	
2021	0	1	0	1	2	0	0	0	0	4
2022	2	1	3	1	0	0	0	1	3	11
2023	1	1	1	0	2	1	0	0	0	6

Note: data provided by the Institute of Virology, Vaccines, and Sera “Torlak”.

**Table 5 vaccines-13-00652-t005:** Vaccines were indicated by volunteers in the study.

Infection	N	Vaccine, % (95% CI)			
		Priorix	M-M-R II	Other	Name Unknown
Measles	112	3.6 (1.4–8.8)	15.2 (9.1–23.0)	8.0 (4.3–14.6)	73.2 (64.3–80.5)
Rubella	85	5.9 (2.5–13.0)	12.9 (7.4–21.7)	9.4 (4.8–17.5)	71.8 (61.4–80.2)
Mumps	91	5.5 (2.4–12.2)	13.2 (7.7–21.6)	9.9 (5.3–17.7)	71.4 (61.4–79.7)

Note: N—number of persons with documented vaccination information.

## Data Availability

Data are contained within the article.

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
