# Peer review of "Herd Immunity to the Measles, Mumps and Rubella Viruses Among the Belgradian Population in May, 2024"

_vaccines, 2025, doi:10.3390/vaccines13060652_

Round 1
Reviewer 1 Report
Comments and Suggestions for Authors
Article: Heard Immunity to the Measles, Mumps and Rubella Viruses Among the Belgradian Population in May 2024.
In this manuscript, Popova et al. analyze seroprevalence of measles, mumps, and rubella IgG (as well as levels) by ELISA in a cross-sectional randomized study. While the study is needed (considering numbers cited for increasing measles cases), the study analysis and manuscript writing has multiple flaws, but this reviewer thinks that with some reanalysis and rewrites, the manuscript could be resubmitted for possible publication.
General comments:
-How were the IgG values for each virus of positive, negative, and inclusive determined? Were they based on negative controls? Prior studies/reports? Or other?
-Throughout the document, there are times where the authors speculate or make assumptions (some particular instances are in the specific comments section). This needs to be removed. Hypothesize or support with citations, only report results in the results section, and in the discussion discuss possibilities based on the data, past data, and limitations of the study.
-The key limitation of the study (which results in problems with how the authors present their data), is that only a small portion of participants could confirm vaccination with medical documentation. While the authors acknowledge insufficient documentation (and how the data conflict with what many people reported), and in this type of study it is nearly impossible to get medical documentation for every infection and vaccination, the participants should be analyzed in a different way (such as confirmed vaccination and other).
-Specific example: Lines 279+/Figure 4: If the authors report insufficient reliability of the information volunteers could provide, then how reviewers and readers be sure the data is reliable. This reviewer accepts the vaccinated group as providing documentation. The remaining should be in other (and include all) if there is no reliability from the patients, especially with the data presented in 287-290, This is also in lines 368-370.
Specific Comments:
-Line 20: remove “anti-“ so that it does not sound like people were against the measles vaccine
-Line 73: Jan-Dec is all of the year, but line 74 makes it sound like it should be a shorter portion of the year. This part of the sentence should be clarified or changed to 2023.
-Line 104: No assumptions should be made (can be a hypothesis or supported by data)
-Line 426: Remove the term “obviously”
Author Response
All responses are in the attached Word file

Reviewer 2 Report
Comments and Suggestions for Authors
The manuscript describe a cross-sectional study of herd immunity to the measles, mumps and rubella viruses among the belgradian population. In the investigation data management and data analysis was scientific and the conclusion was valid. This article examines the herd immunity levels of the Belgrade population against measles, mumps, and rubella viruses in May 2024, revealing differences in seroprevalence among different age groups, and highlighting the need to strengthen constant surveillance and revaccination of vulnerable/seronegative groups. I agree to accept after minor revisions (corrections to text editing). This research design provides basic data and possible hypotheses for future research.
The English could be improved to more clearly express the research.
Comments on the Quality of English LanguageThe English could be improved to more clearly express the research.
Author Response
All responses are in attached Word file

Reviewer 3 Report
Comments and Suggestions for Authors
There needs to be a discussion of biases in selecting (selection bias) the study population. The percent of men vs women (table 2) is large, for example. If MMR vaccine was administered, there does appear to be a difference in seroprevalence among the endpoints (Measles, Mumps and Rubella). Is some "effectiveness" of the MMR this variable? This could be a consequence of the selection population. Certainly, the population studied are the data and results obtained and can be unique to one region. I would include a discussion of this "potential" bias along with the varying effectiveness of this vaccine for the different viruses (Measles, Mumps and Rubella).
Some of the sentence structures can be modified.
Examples:
line 488 ",are more likely to take part in studies of this kind" to take part in these studies
line 490 - "as its sample size was rather small." due to sample size.
494 - change this to these data could not...
A good read-over the paper for these modifications would be helpful.
This is overall a minor English comment.
Author Response

(The authors gave the same response as above.)

Round 2
Reviewer 1 Report
Comments and Suggestions for Authors
The authors have completed the revisions to the best of their ability (it is not possible to obtain medical records for each person).